# A Sensitive Frequency Range Method Based on Laser Ultrasounds for Micro-Crack Depth Determination

**DOI:** 10.3390/s22197221

**Published:** 2022-09-23

**Authors:** Haiyang Li, Wenxin Jiang, Jin Deng, Ruien Yu, Qianghua Pan

**Affiliations:** 1Key Laboratory of Advanced Manufacturing Technology, North University of China, Taiyuan 030051, China; 2China Special Equipment Inspection and Research Institute, Beijing 100029, China

**Keywords:** laser ultrasonics, surface crack, Rayleigh wave, sum of transmission coefficients

## Abstract

The laser ultrasonic method using the characteristics of transmitted Rayleigh waves in the frequency domain to determine micro-crack depth is proposed. A low-pass filter model based on the interaction between Rayleigh waves and surface cracks is built and shows that the stop band, called the sensitive frequency range, is sensitive to the depth of surface cracks. The sum of transmission coefficients in the sensitive frequency range is defined as an evaluated parameter to determine crack depth. Moreover, the effects of the sensitive frequency range and measured distance on the evaluated results are analyzed by the finite-element method to validate the robustness of this depth-evaluating method. The estimated results of surface cracks with depths ranging from 0.08 mm to ~0.5 mm on the FEM models and aluminum-alloy samples demonstrate that the laser ultrasounds using the characteristics of Rayleigh waves in the frequency domain do work for quantitative crack depth.

## 1. Introduction

Randomly distributed surface cracks are difficult to detect and affect the capabilities of the materials due to their gradual expansion, which deepens with service time. The crack depth is a key parameter to evaluate the mechanical properties of materials. The ultrasonic-detection method has the advantages of non-pollution, available wave types, and abundant acoustic phenomenon, and the Rayleigh wave in particular is very sensitive to surface cracks, making it a good tool to quantitatively test the surface crack’s depth [1,2]. However, the coupling between the transducer and the tested material limits the application of the traditional Rayleigh-wave method in severe environments such as high temperature, high pressure, and high corrosion to accomplish quantitative testing of surface cracks. Air-coupled acoustics [3], electromagnetic ultrasonics [4], and laser ultrasonics [5] are the main three non-contact technologies that have gained a lot of attentions. Air-coupled acoustic has a better testing ability in the low-frequency domain because the generation of the acoustic wave with high frequency increases the complexity and difficulty of the detection system [6]. Electromagnetic ultrasonics, which makes use of the electromagnetic phenomenon in the metal material to generate the acoustic wave, is only suitable for detecting the crack in the metal material.

Laser ultrasonic technology without any limitation of the frequency and the tested material is a high-spatial-resolution and non-contact technology that has developed into an important branch of optics and acoustics. In particular, thanks to the scattering pattern of waves at the surface crack containing lots of size information on cracks, tools based on laser ultrasounds are useful and promising for quantitatively determining crack depth [7,8,9].

Linking characteristics of transmitted and reflected waves in the time or frequency domain and crack dimensions have been widely analyzed by many researchers in many publications. Cooper [10] first compared the frequency spectrum of the incident wave in a bandwidth of 10 KHz to 5 MHz with that of reflected and transmitted waves at the crack and pointed out the amplitude reduction in the frequency spectrum related to the wavelength of the incident Rayleigh wave. After that, Hévin [11] presented that a surface crack was viewed as a low-pass filter when Rayleigh waves interacted with the surface crack and proposed spectral ratios to evaluate crack depth by applying spectral analysis of incident and transmitted waves. Kim [12] investigated the A-scan and C-scan ultrasonic-testing data of aluminum alloy, hot-rolled steel, stainless steel, and copper alloy by the laser ultrasonic method. The acquired data indicated the detectable depth for metals via the laser ultrasonic technique. The studies using reflection and transmission coefficients [13,14,15,16] relied on the frequency-spectrum method on account that the amplitude of reflected and transmitted waves must be extracted in the frequency spectrum.

Another method making use of the time lag of transmitted Rayleigh waves or the arrival time of scattered waves at a crack is called the time-domain method. Cooper [10] also presented a time-domain theoretical-calculation equation of crack depth based on the Rayleigh-wave propagation path at the crack. Jian [17] achieved crack-depth gauging with the arrival time of scattered waves at cracks. Ochiai [18] studied a laser ultrasonic system fabricated in conjunction with a time-of-flight method that used propagation characteristics of the ultrasonic diffraction pattern of the crack tip and the diffraction-based crack size in the time domain to gauge an artificial slot with a depth from 0 mm to 10 mm. Shan [19] used the arrival time of longitudinal waves converted at the crack tip to realize crack-depth measurement.

Based on the amount of previous research, Hess [20] and Masserey [21] presented a review of crack-depth measurement methods using both the time and frequency domains and concluded that the measurement method in the time domain does work in situations in which the ratio of wavelength to crack depth is less than 0.4, where the measurement method in the frequency-spectrum domain cannot work, and the measurement method in the frequency domain does work at a ratio of wavelength to crack depth of more than 0.4. To acquire the precise amplitude of reflected and transmitted Rayleigh waves, the detection position is far from the crack to avoid the overlap of the direct waves and reflected waves or the effect of faces of cracks. The methods above are also called far-field measurement methods. On the contrary, Scanning Laser Source (SLS) evaluates surface-crack size by analyzing the amplitude–amplitude of an acoustic signal received by a laser at the fixed position when another laser is illuminated just at and far away from a crack. This technique belongs to the near-field method due to the incident laser only being placed at cracks to monitor the changes in the generated ultrasonic signal. Therefore, SLS achieves higher signal noise than the method above using the scattering pattern of Rayleigh waves and even does work for micro-surface cracks whose depth is less than the wavelength of the incident wave-center frequency. Kromine [22] experimentally validated the possibility of the SLS method to detect small surface-breaking defects in the real structure of turbine disk. Close to the real situation, Arias [23] proposed a thermoelastic model considering the effects of thermal diffusion and constraints such as the width of the laser source. 

More comprehensively, Sohn [24] proposed a hybrid model combining a mass-spring-lattice method and a finite-difference method to simulate the elastic-line source of the SLS technique. Dixon [25] studied the theoretical mechanism of the SLS technique and concluded that the enhancement of Rayleigh waves observed when the crack is irradiated by a laser beam is due to a combination of source truncation, free boundary conditions at the crack edge, and interference effects. So far, both the far-field and the near-field approaches to size surface-track depth are listed. Besides the approaches above, another known as the nonlinear modulation method has been proposed [26] as a novel method using three lasers, one of which is a pumped laser to generate surface Rayleigh waves, another one of which is a probe laser to receive acoustic signals, and the last one of which is to heat cracks to open and close crack surfaces. Thanks to one more laser irradiated at the crack to generate the nonlinear frequency mixing, the nonlinear modulation method is more appropriate for partially closed cracks; nevertheless, other methods are useful for surface-breaking cracks. It is similar to SLS in that the crack also has to be illuminated by using the laser. However, with both the SLS and the nonlinear modulation method, the potential harm caused by the heat of laser illuminating on the surface crack to speed up the expansion of cracks is ignored.

Compared with the methods using laser impacting on cracks, measurement methods using the scattering pattern of Rayleigh waves at cracks do not cause the second damage at the original crack. With the development of artificial-intelligence technology, many researchers [27,28,29,30] have tried to use it to facilitate nondestructive-testing technology based on laser ultrasound. Although this field has great potential for development, the application scenarios and accuracy rates of artificial-intelligence methods such as machine learning are still relatively limited. Therefore, deeper research on the reflection and transmission of Rayleigh waves is still a meaningful subject in the field of evaluating crack depth by laser ultrasounds.

Above all, most studies have measured large surface cracks with laser wavelengths larger than the generated Rayleigh ones. For small surface cracks with depths less than the incident Rayleigh wavelength, the measurement method of spatial characteristics is no longer effective. Therefore, the quantitative detection of micro-surface cracks by the laser ultrasonic Rayleigh-wave technique opens a broad scope of research. The main context of this work includes four parts. The first part is the theoretical analysis of the interaction between the transmitted wave and the surface crack in the frequency domain. The analytical-detection parameter based on the low-pass-filter model is presented in this part. Then, the finite-element models based on the actual size of the experimental samples are simulated to verify the feasibility of the proposed detection method. The setup parameters and simulation process by FEM are presented in the second part. Last, the experimental platform based on laser ultrasonics is built and the quantitative detection for the surface cracks on the aluminum-alloy samples are accomplished. The experimental results and error analysis are detailed in the third part. Finally, the main conclusions are discussed in the fourth part.

## 2. Materials and Methods

The acoustic field at the crack includes the transmitted and reflected acoustic waves generated by the interaction between the surface crack and the Rayleigh wave. These acoustic waves contain a large amount of information on crack size. A rectangular surface crack leading to discontinuing interfaces in the material was used in this work to analyze the transmission waves in the frequency domain. The direct and reflected waves were obtained at the front side of crack interfaces. The transmitted waves were obtained at the back side of crack interfaces. The scattering waves with wave-type transformation at the crack tip were also an important phenomenon. However, the micro-crack depth was so small that scattering waves could be omitted. In this work, the ratio of crack depth to width was 0.4–2.5. The transmission phenomenon of Rayleigh waves with different frequencies at cracks is shown in Figure 1.

Rayleigh waves generated by laser ultrasonics have a broad range of frequencies, so they appear as a summation signal made up of single-frequency signals at different frequencies. By the way of plane-wave decomposition, the incident wave propagates along the x-axis direction, where *A_i_* is the amplitude of the frequency component *w_i_*. The number *n* is decided by the frequency bandwidth of the incident wave. Ignoring the propagation item, vector *A* = [*A*_1_*^in^*, *A*_2_*^in^*, *A*_3_*^in^..A_i_^in^*,*...*, *A_n_^in^*] is made up of individual frequency components’ amplitudes, vector *T =* [*T*_1_(*w_1_*), *T*_2_(*w_2_*), *T*_3_(*w_3_*)*...*, *T_i_*(*w_i_*)*...T_n_*(*w_n_*)] is transmission coefficients of the corresponding frequency components, and vector *W =* [*exp*(*jw_1_t*), *exp*(*jw_2_t*), *exp*(*jw_3_t*),*...*, *exp*(*jw_i_t*),*...*, *exp*(*jw_n_t*)] is frequency components in the frequency band of the Rayleigh wave generated by the laser. Due to that, each component in the broad frequency range has different transmission capability, so the transmission coefficient *T_i_* is related to the value of frequency *w_i_*. For Rayleigh-wave transmission at the surface crack, the crack is equal to a low-pass filter with a cutoff frequency that is relative to the crack’s depth. It was assumed that the relationship between transmission coefficients and frequency values is approximately *T_i_*(*w_i_*) *= k* × *w_i_ + b*, where *k* denotes the transmission ability decrease with the frequencies and *b* is relevant to the crack depth. The transmitted Rayleigh wave at the surface crack is described as
(1)$$\begin{array}{c} A^{t}e^{jwt}=\left[A_{1}^{in},A_{2}^{in},A_{3}^{in},\ldots ,A_{c}^{in},\ldots ,A_{n}^{in}\right]\left[\begin{array}{ccccccc} 1 & & & & & & \\ & 1 & & & & & \\ & & 1 & & & & \\ & & & \ddots & & & \\ & & & & {-k}_{c}w_{c}+b & & \\ & & & & & \ddots & \\ & & & & & & -k_{n}w_{n}+b \end{array}\right]\left[\begin{array}{c} e^{jw_{1}t}\\ e^{jw_{2}t}\\ e^{jw_{3}t}\\ \begin{array}{c} \vdots \\ e^{jw_{c}t}\\ \vdots \end{array}\\ e^{jw_{n}t} \end{array}\right]=\sum_{c}^{n}A_{i}^{in}\cdot \left(k_{i}w_{i}\right)\cdot e^{jw_{i}t}\;\# \end{array}$$
where *T_c_ = k_c_w_c_ + b* is the transmission coefficient at the cut-off frequency of the low-pass filter. The low-pass filter in Equation (1) is an ideal filter with transmission coefficients equal to one when the incident frequencies are lower than the cut-off frequency. This means that the frequency components in the range of 0~*w_c_* pass through the surface crack without any loss. Therefore, the surface crack is seen as an ideal low-pass filter by Equation (1). To analyze Equation (1) in detail, assuming that the frequency band ranges from 0 to ~5 MHz, the low-pass filters under the condition that the value *k* is −1 and b ranges from 4 to 4.5.

The reason that *k* was set as −1 is to describe the relationship between the transmission ability and the frequency of individual components as a negative correlation. This is consistent with the transmission ability of the components in the frequency band. The *b* in Equation (1), denoting the transmission ability of the Rayleigh wave, is related to the depth of the surface crack. It is seen in Figure 2 that under the different values of b the low-pass filters had similar properties in two parts, which are the passband from 0 to *w_c_* and the stopband from *w_c_* to 5 MHz, forming a low-pass filter. In the passband, the transmission coefficients were 1 for all the low-pass filters, leading to overlap in this frequency range. In the stopband, the transmission coefficients increased with the values of *b* in Equation (1) and decreased with the frequencies. The physical meaning of the value of *b* in Equation (1) is crucial. At the same frequency in the stopband, the bigger the value of *b* is the stronger the transmission ability is. It is known that if the depth of the surface crack is deeper it leads to weaker transmission ability of the Rayleigh wave. That is to say, a negative-correlation relationship exists between the value of *b* and the depth of the surface crack. Therefore, the transmission-modulation model of the surface crack on the wave propagation was built thanks to Equation (1).

Moreover, it is worth noting that the transmission coefficients of components in the stopband were more sensitive to the depth of the surface crack than those in the passband, according to Figure 2. The stopband of the ideal low-pass filter is named the sensitive-frequency range, where the ability of transmission waves is relevant to the size of the surface cracks. Furthermore, the transmitted wave is decomposed by the plane-wave-decomposition method, and a characteristic parameter of the sensitive-frequency range is obtained as
(2)$$T=\overset{n}{\underset{c}{\sum }}A_{i}^{t}\cdot e^{jw_{i}t}=\overset{n}{\underset{c}{\sum }}A_{i}^{in}\cdot T_{i}\cdot e^{jw_{i}t}=\overset{n}{\underset{c}{\sum }}T_{i}$$

It is seen from Equation (2) that a characteristic parameter that is relevant to crack depth in the frequency domain is finally obtained. This characteristic parameter is a sum form calculated in the sensitive-frequency range. Therefore, the characteristic parameter defined by Equation (2) has the potential to evaluate crack depth. The relationship between the value of *b* and the normalized sum of transmission coefficients in the sensitive frequency range was calculated by Equation (2) and the simulation results are shown in Figure 3.

It is seen that there was a linear relationship between the value of *b* and the normalized sum of transmission coefficients, demonstrating that the characteristic parameter based on Equation (2) proposed in this work has the potential to quantify the depth of the surface crack. On account of the negative-correlation relationship between the value of *b* and the depth of the surface crack, the results in Figure 3 are presented using the descending order of the x-axis to conveniently compare the simulation and experimental results in the next sections.

## 3. Numerical Simulation

### 3.1. Finite-Element Model

The finite-element method (FEM) can flexibly handle complex geometric models and obtain full-field numerical solutions. It has been widely used to calculate the generation and propagation of laser ultrasonic waves in materials. In this work, the finite-element method by COMSOL Multiphysics was used to simulate the physical interaction of surface Rayleigh waves excited by a pulsed-laser line source and the surface cracks of an aluminum plate. First of all, the transient-temperature field distribution of the material model was calculated according to the heat-conduction equation, and then the transient-temperature field was used as the body load to solve for the displacement field. The heat-conduction equation and finite-element-control equation can be expressed as:(3)$$K_{T}T+C_{P}\dot{T}=p_{1}+p_{2}$$
(4)$$M\ddot{U}+KU=F_{ext}$$

In Equation (3), $K_{T\;}$ is the heat-conduction matrix, $C_{P}$ is the heat-capacity matrix, *T* is the temperature vector and $\dot{T}$ is the temperature-rise-rate vector, $p_{1}\;$is the heat-flux vector, and $p_{2}\;$is the heat-source vector. For wave propagation, ignoring damping, Equation (4) is the governing finite-element equation, where *M* is the mass matrix, *K* is the stiffness matrix, *U* is the displacement vector, and $\ddot{U}$ is the acceleration vector. For thermoelastic bodies, the external-force vector for an element $F_{ext}$ can be calculated by the following equation:(5)$$F_{ext}=\int_{Ve}B^{T}TD\varepsilon_{0}dV$$

In Equation (5), Ve is the volume of the element, $B^{T}$ is the transpose of the derivative of the shape functions, D is the material matrix, and $\varepsilon_{0}$ is the thermal-strain vector. The detailed physical parameters of the model are shown in Table 1 below.

The accuracy of the FEM for simulating laser ultrasound is determined by the time step and mesh size, which are derived by calculating the maximum frequency during the simulation of FEM. The maximum frequency of laser-excited ultrasound can be calculated from Equation (6):(6)$$f_{max}=\frac{\sqrt{2}C}{\pi r_{0}}$$
where the $r_{0}$ is the radius of the pulsed-laser source, which was set as 0.1 mm in this work, and the C is the Rayleigh-wave velocity of 2990 m/s. The time step was set as 15 ns based on $\Delta_{t}=1/\left(20f_{max}\right)$. The spatial grid used triangles with a size of 0.6 μm–0.3 mm. Rayleigh waves, whose propagation orientation is vertical to the source, were generated by the line source that was parallel to one side of the sample in this work. A two-dimensional model was built based on the above simulation parameters, and a product of two Gaussian functions in the time and space domain was used to describe the excitation of the laser source with a power of 0.6 MW and a rise time of 6 ns. The FEM simulations were performed on models of six different depths ranging from 0.08 to ~0.5 mm with a width of 0.2 mm, as shown in Table 2. The crack sizes presented in Table 2 are consistent with those of the experimental samples in the following sections to compare the simulation and experimental results and verify the analytical model in Section 2. The depths of these surface cracks were less than the wavelengths of the laser ultrasonics during the FEM simulation and experimental process. This is discussed in Section 3.2 and Section 4.2.

### 3.2. Analysis of Acoustic Field at the Crack

Based on the FEM setup in Section 3.1, the simulation-displacement field at 4.35 μs was obtained and is presented in Figure 4a, where the head wave, longitudinal wave, transverse wave, and Rayleigh wave were simultaneously generated by the incident laser impacting on the surface of the simulation model. In particular, the reflected wave, transmitted wave, and scattering acoustic wave were also observed at the defect on account of the Rayleigh wave meeting the defect during its propagation process. In Figure 4, the distance between the laser source and the crack was 9 mm. Two probes named A and B, which were 2 mm and 3 mm away from the crack, respectively, were placed on the left and right sides of the crack, respectively, to acquire the reflected and transmitted waves.

In Figure 4b,c the acoustic wave labeled in the blue color was received at probe A and the acoustic wave labeled in the red color was received at probe B. The distance between the excitation point with probe A was 7 mm, so the arrival time of the direct Rayleigh wave was 2.3 μs based on the theoretical calculation, which is the same as the 2.3 μs indicated in Figure 4b. The arrival time of the reflected wave was 3.7 μs, which is consistent with the wave-propagation distance of 11 mm in the Figure 4b. To expose the influence of the surface crack on the frequency spectrum of transmitted waves, the frequency spectrums of transmitted waves and direct waves by FFT were compared and are shown in Figure 4b, where it can be seen that the amplitude of the transmitted wave had an obvious attenuation in the specific frequency range. The center frequency of the direct wave was 2.6 MHz and the center frequency of the transmitted wave was 1.56 MHz. A center-frequency shift happened between the direct and transmitted waves. This phenomenon can be explained using a power function that expounds the modulation effect of the surface crack on the frequency spectrum of a transmitted wave due to the transmitted ability difference of each frequency component [14]. The specific frequency component, which had a bigger attenuation of transmitted amplitude at the surface crack, was more sensitive to the depth of the crack. Besides, the wavelength of the Rayleigh wave was calculated by the center frequency of the direct wave as 1.1 mm. The maximum surface-crack depth was 0.5 mm, as shown in Table 2, which is less than the wavelength, so all the surface cracks designed in this work belonged to the category of micro cracks. To determine the sensitive range in the frequency spectrum, the transmission coefficients at the crack with the depths shown in Table 2 are presented in Figure 5.

In Figure 5b the calculated transmission coefficients are shown as a function of frequency for six depths of cracks. In the low-frequency range, the waves had a greater ability to go through the crack without much energy loss. In Figure 5b, approximately below the center frequency of the transmitted wave, the amplitude of the direct wave and the transmitted wave was almost the same. Corresponding to this phenomenon, the transmitted coefficients in the frequency domain were almost the same under the different crack depths shown in Figure 5a. In this situation, the transmitted coefficient cannot be used as a characteristic parameter to evaluate crack depth. A red box is labeled in Figure 5a to expose the sensitive-frequency range to crack depth. The frequency range in the red box was from 1.0 MHz to 2.6 MHz. For frequencies such as 0.26 MHz, 0.52 MHz, 3.65 MHz, and 3.91 MHz out of the frequency band of 1.0 MHz~2.6 MHz, the transmitted coefficients versus the crack depths are shown in Figure 5b. These curves were jumbled together, leading to the failure of crack-depth measurement. for frequencies such as 1.04 MHz and 1.30 MHz in the frequency band of 1.0 MHz–2.6 MHz, the transmitted coefficients versus the crack depths are shown in Figure 5c. Compared to the curves in Figure 5b, the curves at the sensitive frequencies were orderly and presented with a similar trend as the transmitted coefficients decreased as crack depth increased. Meanwhile, at the same crack depth, the transmitted coefficients decreased as the frequencies increased. The rules in the sensitive-frequency band related to the crack depth present the possibility to evaluate crack depths. Therefore, the sensitive-frequency range in Equation (1) was from 1.0 MHz to 2.6 MHz. Then it was suitable for the sum of the transmission coefficients to be calculated in this frequency range to evaluate crack depth.

### 3.3. The Effect of Excitation Conditions on the Sensitive-Frequency Band

To determine the effect of three main condition parameters of the incident laser source (pulse time, power, and radius) on the frequency spectrum of the transmitted wave, six simulation results were achieved by changing only one of the above three simulation parameters at a time, and the results are shown in Figure 6.

Figure 6 represents the changes of the transmitted wave in the frequency domain when the parameters of the FEM model were changed. Comparing Figure 6a with Figure 6b, and Figure 6c with Figure 6d, it is obvious that the amplitude of the transmitted signal in Figure 6b was larger than that in Figure 6a and the amplitude of the transmitted signal in Figure 6d was also larger than that in Figure 6c. The reason for this phenomenon is that the stronger laser source releases more energy into the material, leading to a bigger amplitude of transmitted waves being generated. The increase in pulse time and power of the laser source converts the higher energy of the incident laser into the mechanical energy of acoustic waves. Moreover, the bandwidths of transmitted frequency also increased, as shown in Figure 6b,d, with the increase in pulse time and power of the laser source. The amplitude of all frequency components was enhanced by the stronger acoustic wave generated by the laser, leading to the frequency bandwidth being wider, and after that the weak-amplitude component, which did not affect the bandwidth, was enhanced. In contrast, the radius of the laser source increased, whereas the other excitation conditions of the liner laser were unchanged, and the lesser energy of the laser in the unit area was irradiated into the material, resulting in the amplitude of the transmitted waves decreasing. On account of the decrease in energy of the acoustic waves generated by a laser source, the frequency bandwidth of transmitted waves was narrower with increasing radius of the laser source. The frequency spectrum of transmitted waves is shown in Figure 6e,f. Furthermore, the transmission coefficients under these excitation conditions were calculated by Equation (2) and are shown in Figure 7.

The results in Figure 7 present the influence of the excitation conditions of the incident laser on the transmitted coefficients in the frequency domain. In Section 3.1, the sensitive frequency range of the transmitted wave was 1 MHz–2.6 MHz. The repetition of determining the sensitive-frequency band under the different excitation conditions shows that the sensitive-frequency band is only related to the crack depth. The amplitude and frequency bandwidth of the frequency spectrum of transmitted waves under three different excitation conditions of laser radiation were analyzed and are presented in Figure 7. It is worth noting that not only did the amplitude and frequency bandwidth of the frequency spectrum of transmitted waves change under the different excitation conditions of the laser source, but the direct wave also showed the same trend. The transmitted coefficient was defined using Equation (1) as the ratio of the amplitude of the transmitted wave and the amplitude of the direct wave. The definition of the transmitted coefficient led to the transmitted coefficients being unchanged. Therefore, the sensitive frequency ranges in Figure 7 were unchanged even though the excitation conditions of the laser source changed. It was seen that the characteristic parameter in the sensitive frequency could suppress and eliminate the measurement errors generated by the differences due to multiple different operations.

## 4. Experimental Setup and Result

### 4.1. Experimental Setup

The schematic block diagram of the experimental system is shown in Figure 8. The red and green lines in Figure 8 are the transmitting and receiving laser paths, respectively. A laser was generated by the laser transmitter of the CFR 2000 type and passed through a lens with a focal length of 100 mm focused on a line source on the surface of the specimen. The acoustic field, including the head, transverse, longitude, and Rayleigh waves, were excited at the laser source. After the interaction between the Rayleigh wave and the surface crack, the acoustic signals carrying the size information of the surface crack were received by the QUARTET-500 mV receiving unit via an interference method. During the scanning process, the transmitting and receiving parts remained fixed and the sample was moved by the motion controller. Finally, the LU Scan software on the computer stored and displayed the acoustic signals.

This work used 6061 aluminum-alloy material to make six samples with different depths of surface cracks. The crack depth of the sample also corresponded to the simulation model in Table 2. The distance between the laser-excitation position and the signal-receiving position was fixed at 15 cm, and the surface scanning was realized by moving the tested samples.

As Figure 9 shows, the scanning area was divided into zones I and II on the specimen surface. Zone I was an area where both the excitation laser and reception laser were located on the left side of the surface crack. In this case, the Rayleigh wave did not pass through the surface crack, and only the direct wave and reflected wave were received. In Zone II, the excitation laser and reception laser were on the left and right sides of the surface crack, respectively. In this situation, the Rayleigh wave mee the surface crack, resulting in the transmitted wave being generated and received by the reception laser. It is worth noting that besides these observed waves, the reflected waves at the edges and bottom of specimens were also received. However, other reflected waves were not helpful for depth measurement of cracks. Therefore, the relationship between surface-crack depth and the transmission coefficient was established by analyzing ultrasonic signals collected in Zone II to quantitatively detect the surface crack on aluminum specimen B.

### 4.2. The Analysis of Experimental Signals

The B-scan patterns are presented in Figure 10. During the scanning process, the reflected, transmitted, and direct waves could be observed, and the location of the surface cracks can be distinguished in the image.

Figure 10 was obtained with a specimen movement distance of 15 cm. The two zones defined in Section 4.1 are also labeled in this image. In Zone I, the received waves, including reflected and direct waves, were collected at the position labeled in the red line in Figure 10b and are shown in Figure 10c. The signals at another position labeled with the blue line in Figure 10b across the whole scanning distance are presented in Figure 10a. According to the B-scan image, the position of the crack was easily recognized at 7.38 cm. In the Figure 10a, the amplitude of the transmitted waves in Zone II was less than the amplitude of direct waves in Zone I due to the existence of the surface crack. The reflected and transmitted acoustic signals in the time domain were chosen at 11.8 cm from the B-scan image and are presented in Figure 11a. The frequency spectrum of the acoustic waves is shown in Figure 11b.

Due to the distance of the excitation position and reception position being fixed at 0.5 cm, the arrival time of the direct wave was theoretically calculated as 17 μs, which is close to the value of 17.2 μs in Figure 11a. The arrival time of the reflected wave was 27.2 μs in the case of the receiver position and was 4.17 cm away from the surface crack because the reception position was at 11.55 cm and the crack was at 7.38 cm. The arrival time of the transmitted wave was 17.2 μs, the same as the arrival time of the direct wave. In Figure 11b, a significant attenuation of the amplitude of the transmitted wave compared to that of the direct wave can be seen in the specific-frequency range. These results are similar to the simulation results in Section 3.2. Using the experimental data in Figure 8, the transmitted coefficients in the frequency domain at the crack depth of 0.08 mm, 0.1 mm, 0.2 mm, 0.3 mm, 0.4 mm, and 0.5 mm are calculated and are shown in Figure 12.

Figure 12 clearly shows the relationship between the crack depth and the transmission coefficient in different frequency ranges. As per the red-highlighted area in the graph, there was a negative linear relationship between frequency values and the transmission coefficient in the range of 1.0–2.6 MHz. The red box shows the sensitive frequency range that was used in Equation (1) to estimate the crack depth. The experimental results agree with the conclusions observed by the FEM shown in Figure 7. In addition, the transmitted coefficients in Figure 7 are the mean value obtained by averaging the three measurement values to get rid of the effect of random errors.

### 4.3. The Contrast between the Experimental and Simulated Results

Considering that acoustic waves have different transmission capabilities in different frequency ranges, the five frequency bands were as follows: (a) 1.0 MHz–2.6 MHz, (b) 1.0 MHz–2.0 MHz, (c) 1.5 MHz–2.6 MHz, (d) 0.5 MHz–2.6 MHz, and (e) 0.5 MHz–3 MHz, where frequency bands (a), (b), and (c) were in the sensitive frequency range and frequency bands (d) and (e) were over the frequency range. The corresponding transmitted coefficients in Figure 13 located at the frequency bands in (a), (b), (c), (d), and (e) were extracted and input into Equations (1) and (2) to calculate the sum of the transmission coefficients in a sensitive-frequency band to evaluate the crack depth. The sum of the transmission coefficients derived from the experiments and simulations was fitted linearly with the crack depth, as shown in Figure 13. The red dots and lines in Figure 13 are the initial data and the fitted lines of the experimental data, respectively. The blue dots and blue lines in Figure 13 are the initial data and the fitted lines of the simulation data, respectively. The fitted curves for the experimental and simulated data are presented in Table 3.

As shown in Figure 13a–c, the simulation results are very consistent with the experimental data, and there was a linear relationship between the crack depths and the normalized parameters, which were normalized by the sum of the transmitted coefficients. Fitted curves D1–D6 in Table 3 correspond to the simulation and experimental data in Figure 13a–c. These fitted curves can be used for evaluating crack depths. The three frequency bands used in Figure 13a–c belong to the sensitive -frequency band shown in Figure 5a and Figure 12. The frequency band used in (a) was just equal to the sensitive-frequency band. The frequency band used in (b) and (c) was in the range of the sensitive-frequency band. The fitted curves in Figure 11a–c presented the same rules between the crack depth and the normalized parameter. The results in Figure 5c show that the relationships between the crack depth and the transmitted coefficients have the same trend in the sensitive-frequency band. Consequently, at the same crack depth, the sum of transmitted coefficients could be used for the depth measurement. Therefore, any frequency band belonging to the sensitive-frequency band was used for crack-depth measurement.

In Figure 13d,e, the fitted curves D7–D10 obtained in the non-sensitive-frequency band were different from the others. Just as expected, fitted curve D7 was inconsistent with fitted curve D8 and fitted curve D9 was inconsistent with fitted curve D10. The reason for this is that the frequency band used in Figure 13d,e involved the non-sensitive-frequency band. The frequency band of (d) in 0.5 MHz–2.6 MHz had two frequency ranges. One of them in 0.5 MHz–1.0 MHz was not sensitive to crack depth, according to Figure 5a and Figure 10. The other one in 1.0 MHz–2.6 MHz was sensitive to crack depth. However, all the transmitted coefficients in the two frequency ranges were input into Equations (4) and (5), resulting in decreasing the accuracy of the crack-depth measurement. The same reason explains why fitted curves D9 and D10 were not suitable for crack-depth measurement. The frequency band (e) in 0.5 MHz–3 MHz was divided into three ranges, which were 0.5 MHz–1 MHz, 1 MHz–2.6 MHz, and 2.6 MHz–3 MHz. The first and last frequency band was relatively insensitive. Compared to frequency band (d), more insensitive-frequency components were included in frequency band (e). Therefore, the linear relationship between the crack depth and normalized parameters disappeared. The worse-fitted curves obtained in frequency band (e) are shown. The randomness of fitted curves D7–D10 is presented due to the involved insensitive-frequency band. This corresponds to the messy curves in Figure 5b. Therefore, more non-sensitive frequency was included in the detected frequency band and more measurement errors were added to the estimated depths. The fitted equations for the curves in Figure 13 are presented in Table 3.

As can be seen in Figure 13, the depths of cracks and the normalized sum of transmission coefficients had a better linear relationship in the frequency range of 1 MHz–2.6 MHz than in other frequency bands. In Table 3, 10 fitted equations for the curves in Figure 13 were obtained. Their coefficients were calculated by the first-order polynomial-fitted method. Meanwhile, the experimental results and FEM results were in better agreement in this frequency range. This proves that the normalized sum of the transmission coefficients mentioned in Section 2 can be used as a characteristic parameter to quantitatively assess the depth of surface cracks. Moreover, the D1–D6 curves fitted in the sensitive-frequency bands had similar fitted equations, as shown in Table 3. Therefore, the rule between the crack depth and the sum of the transmission coefficients is general for surface-crack depth measurement in the sensitive-frequency band. This means that the sum of the transmitted coefficients in the detected frequency band, which belongs to the sensitive-frequency range, can be used as a characteristic parameter to measure the crack depth. Once the non-sensitive-frequency band is involved in the detected-frequency band, the precision of crack-depth measurement is reduced. To verify the equation in the experiment and decrease randomness in the experiment, the method mentioned above was used to analyze different scanning positions.

In Figure 14, the red dots are the mean value obtained at several random positions, the black straight line is fitted equation D1 in the 1.0 MHz–2.6 MHz range, and the blue-and-white squares are the absolute errors between the mean values and the fitted equations. In addition to the error at 0.1 mm being relatively large, the absolute errors of other depths were much smaller than the true value. It can be observed that 0.08 mm and 0.1 mm were similar in size, but the calculated results at 0.08 mm showed good accuracy. This is because the magnitude of the crack depth was very small, and the negative impact of some very small experimental and computational errors on the results was magnified. In practice, these errors can be reduced or eliminated by further increasing the accuracy of the experiment and by executing a large number of repeated experiments. In general, the crack depths calculated by characteristic parameters were close to true values, so it can be summarized that the analytical equation has good accuracy for calculating the depths of surface cracks.

## 5. Conclusions

To estimate the depth of micro-surface cracks using laser ultrasonics, the frequency characteristics of transmitted waves at the surface crack was analyzed using FEM and the experimental method. Starting with the different transmitted abilities of different frequency components, a sum of transmitted coefficients in the sensitive-frequency range was proposed to evaluate the depth of surface cracks. The main conclusions were drawn as follows:(1)The sensitive-frequency range was determined by a low-pass-filter model that was built for analyzing the transmitted interaction between the Rayleigh wave and the surface crack in the frequency domain. An analytic equation based on the sum of transmitted coefficients in the sensitive-frequency range was given to quantitatively measure crack depth.(2)According to the FEM simulation results under three different excitation conditions of the laser source, it was seen that the proposed detection method in this work had good repeatability and robustness due to the sensitive-frequency range of the transmitted wave having nothing to do with the excitation conditions of the laser source and being related only to crack depth.(3)The availability of the estimated parameter proposed in this work to evaluate surface-crack depth was validated by using both FEM and the experimental method performed on the micro-surface cracks with depths of 0.08 mm, 0.1 mm, 0.2 mm, 0.3 mm, 0.4 mm, and 0.5 mm, which were less than the wavelength of the Rayleigh wave.

## Figures and Tables

**Figure 1 sensors-22-07221-f001:**
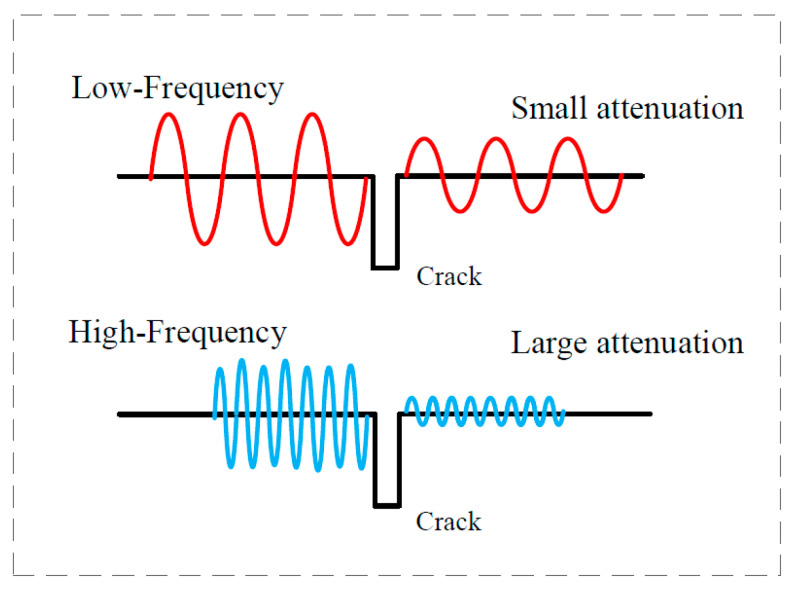
A schematic diagram of signals passing through the crack at different frequencies.

**Figure 2 sensors-22-07221-f002:**
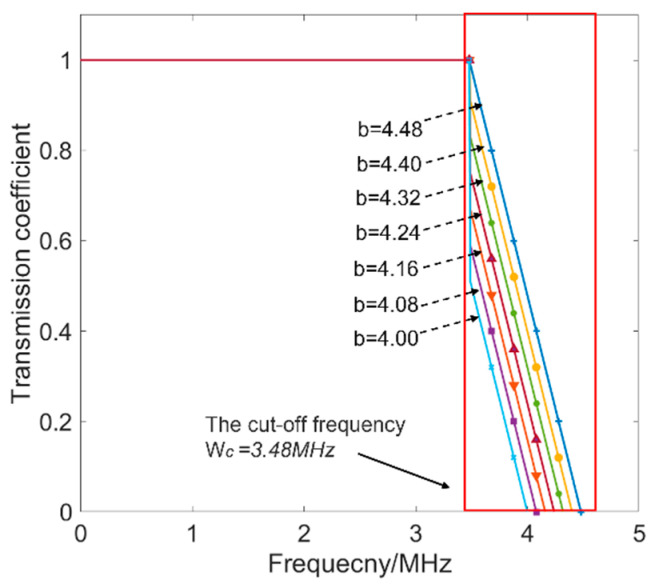
The transmitted coefficients in the frequency band.

**Figure 3 sensors-22-07221-f003:**
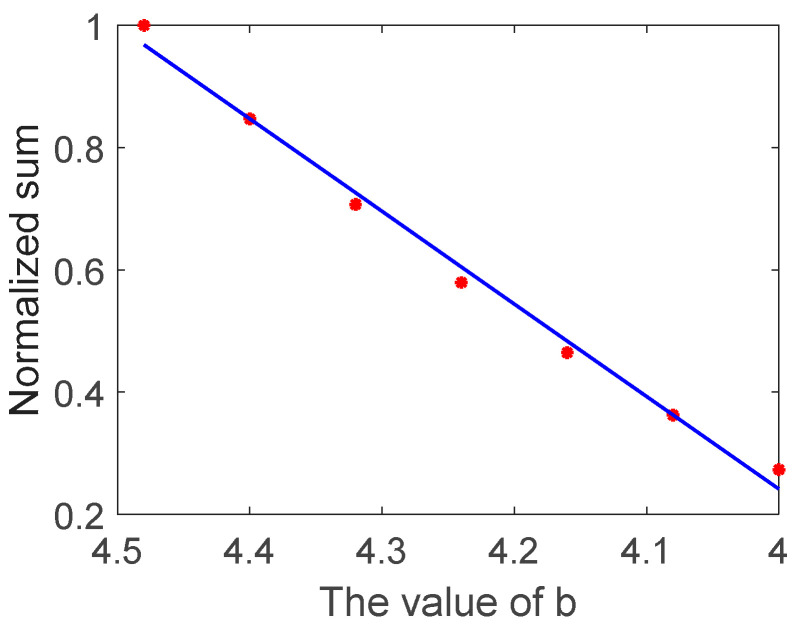
The relationship between the value of *b* and the normalized sum.

**Figure 4 sensors-22-07221-f004:**
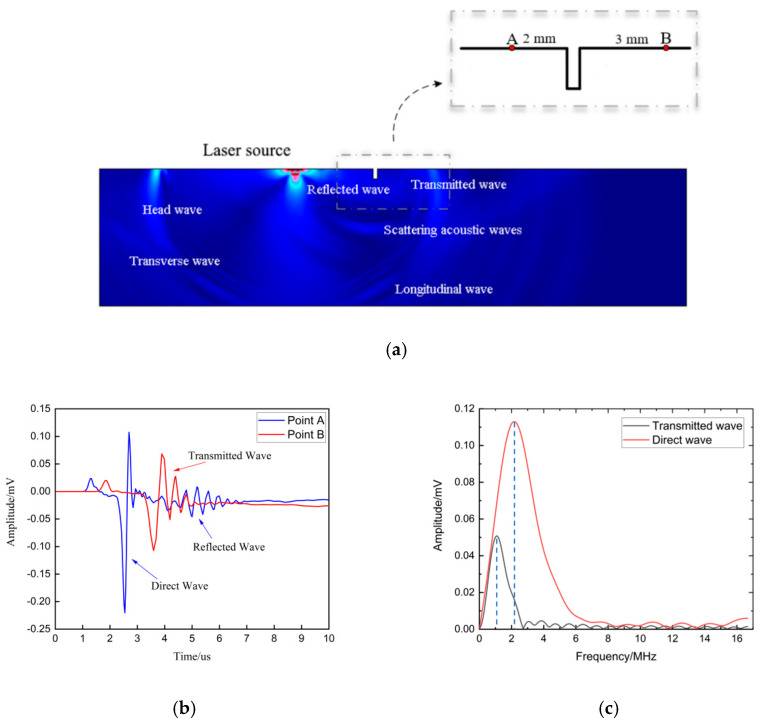
(**a**) Simulation acoustic field; (**b**) acoustic wave in the time domain; (**c**) acoustic wave in the frequency domain.

**Figure 5 sensors-22-07221-f005:**
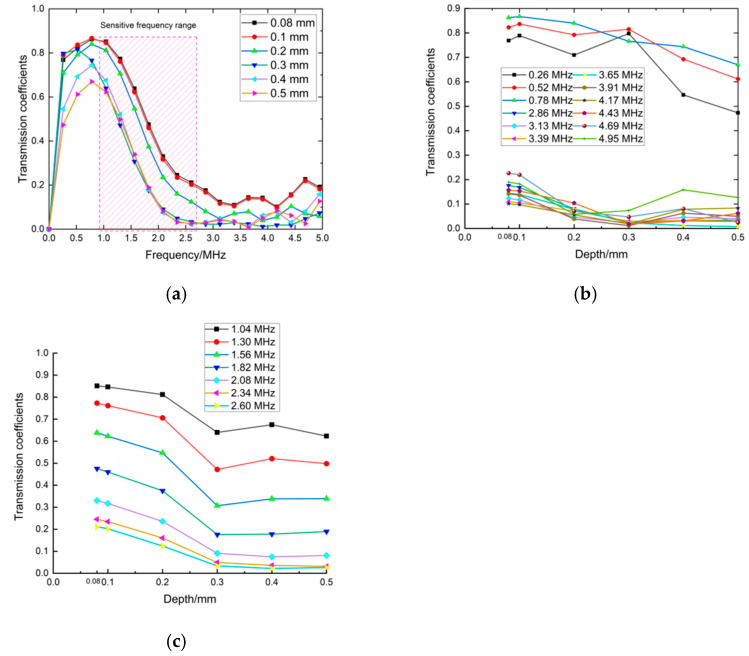
Transmission coefficients under different depths in the whole frequency range in (**a**), in the frequency ranging in 0.26 MHz~0.78 MHz and 2.86 MHz~4.95 MHz in (**b**) and in the frequency ranging in 1.04 MHz~2.60 MHz in (**c**).

**Figure 6 sensors-22-07221-f006:**
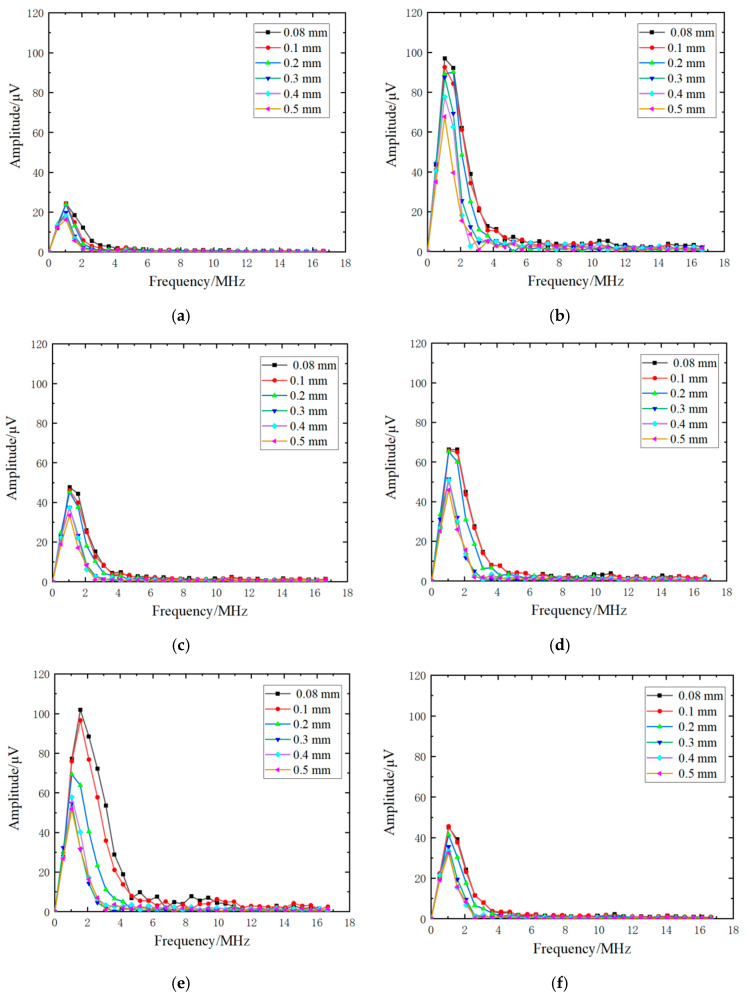
The frequency spectrum of the transmitted wave with a (**a**) pulse-rise time of 3 ns, (**b**) pulse-rise time of 10 ns, (**c**) laser-source power of 0.5 MW, (**d**) laser-source power of 0.7 MW, (**e**) laser-source radius of 80 mm, and (**f**) laser-source radius of 120 mm at the crack with a depth of 0.08 mm, 0.1 mm, 0.2 mm, 0.3 mm, 0.4 mm, and 0.5 mm.

**Figure 7 sensors-22-07221-f007:**
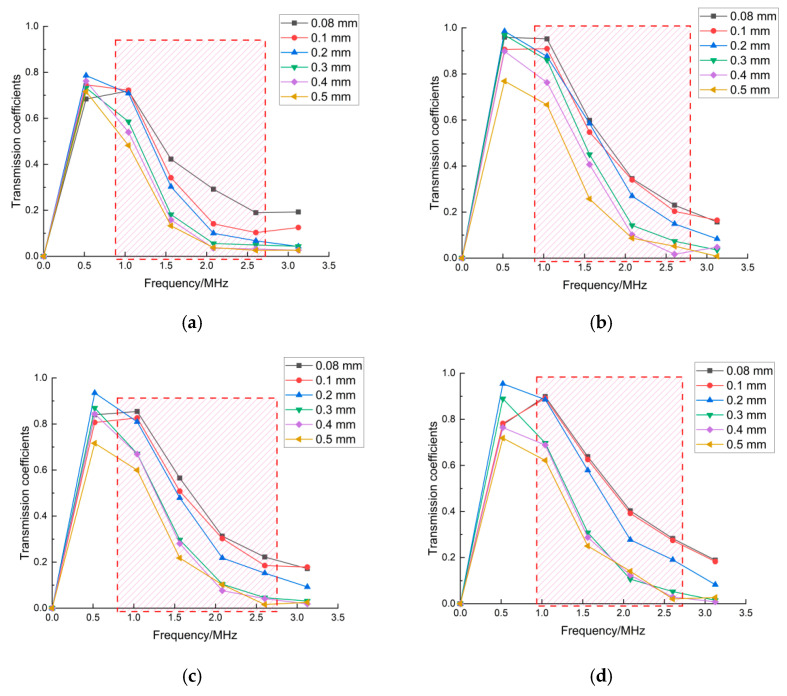
Transmission coefficients: (**a**) pulse-rise time of 3 ns; (**b**) pulse-rise time of 10 ns; (**c**) laser-source power of 0.5 MW; (**d**) laser-source power of 0.7 MW; (**e**) laser-source radius of 80 mm; (**f**) laser-source radius of 120 mm.

**Figure 8 sensors-22-07221-f008:**
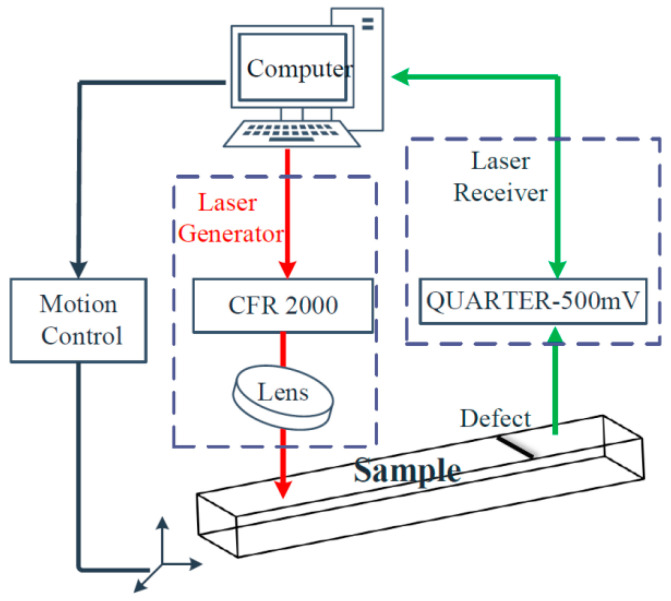
Schematic diagram of the experimental system.

**Figure 9 sensors-22-07221-f009:**
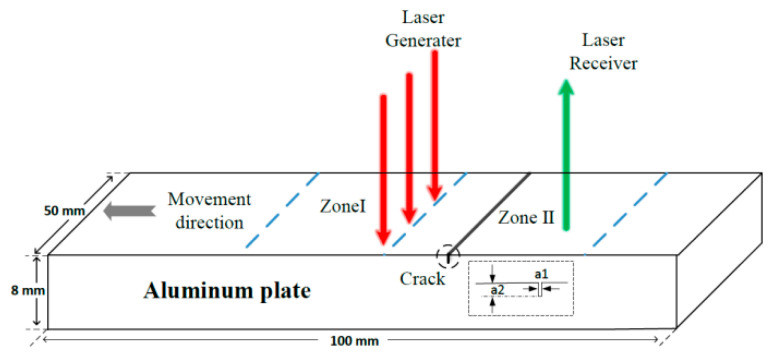
Schematic diagram of scanning-laser-source technology device.

**Figure 10 sensors-22-07221-f010:**
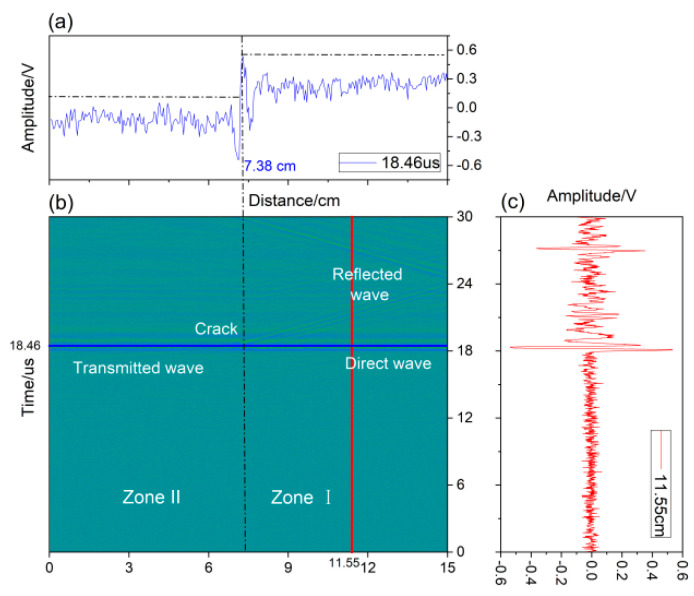
B-scan image.

**Figure 11 sensors-22-07221-f011:**
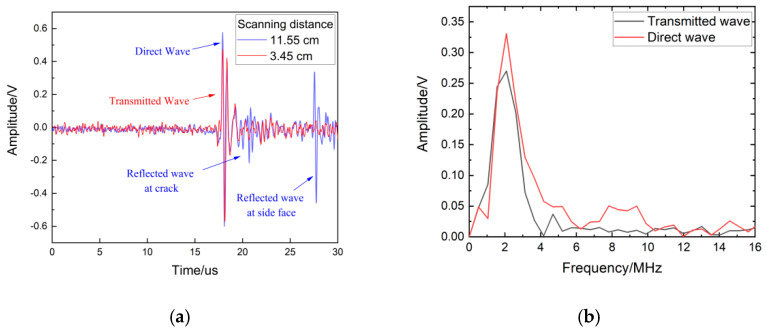
Experimental signals: (**a**) time domain; (**b**) frequency domain.

**Figure 12 sensors-22-07221-f012:**
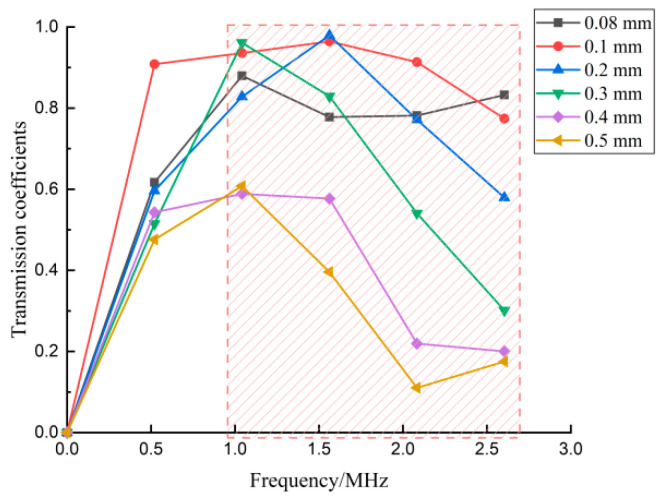
Experimental transmission coefficients.

**Figure 13 sensors-22-07221-f013:**
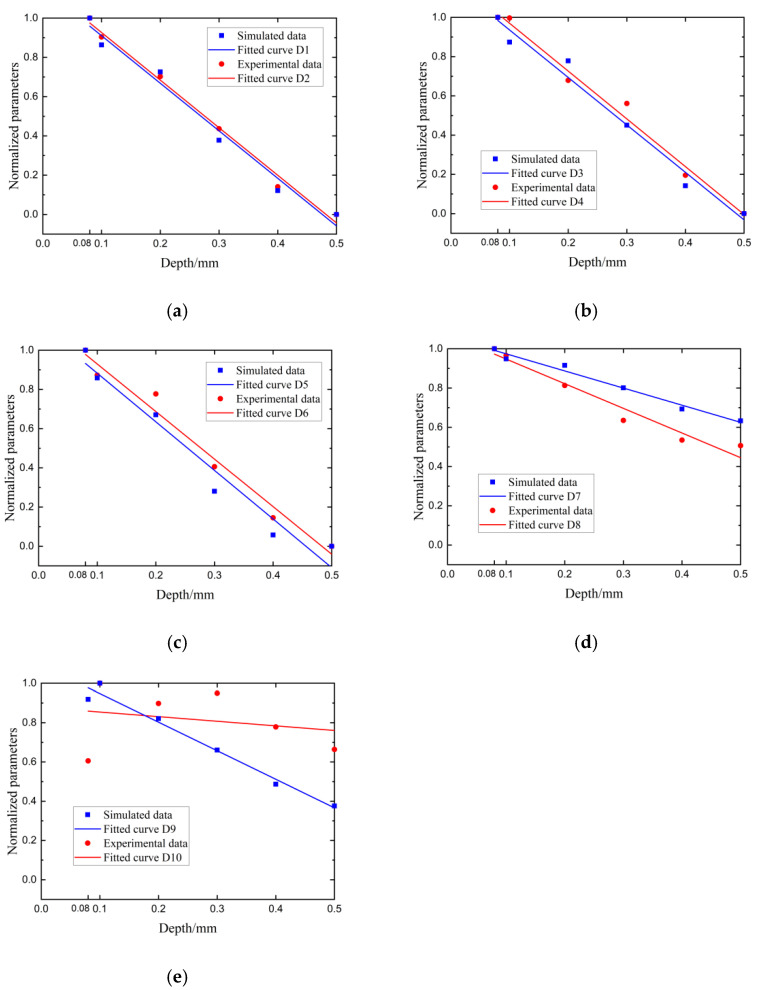
Crack-depth normalized parameters in (**a**) 1.0 MHz–2.6 MHz, (**b**) 1.0 MHz–2.0 MHz, (**c**) 1.5 MHz–2.6 MHz, (**d**) 0.5 MHz–2.6 MHz, and (**e**) 0.5 MHz–3 MHz.

**Figure 14 sensors-22-07221-f014:**
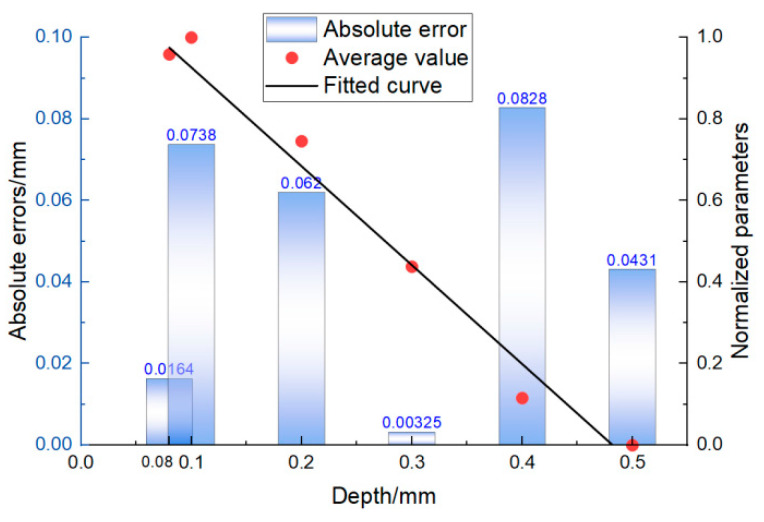
Errors of the fitted equations.

**Table 1 sensors-22-07221-t001:** Physical-parameter settings.

Physical Properties	Value
Constant-pressure heat capacity	900 J/(kg*K)
Thermal conductivity	238 W/(m*K)
Thermal-expansion coefficient	23 × 10^−6^ 1/K
Density	2700 kg/m^3^
Young’s modulus	70 × 10^9^ Pa
Poisson’s ratio	0.33

**Table 2 sensors-22-07221-t002:** The depths of the crack.

Sample Number	Crack Depth/mm
A	0.08
B	0.1
C	0.2
D	0.3
E	0.4
F	0.5

**Table 3 sensors-22-07221-t003:** Fitted equations for the curves in Figure 13.

Experimental-Data Fitted Equation	Simulation-Data Fitted Equation
$D_{1}=-2.42443x\;+1.16909$	$D_{2}=-2.42197\;x+1.15252$
$D_{3}=-2.4362x+1.21372$	$D_{4}=-2.41929x\;+1.17786$
$D_{5}=-2.42534x+1.17222$	$D_{6}=-2.479x+1.13063$
$D_{7}=-1.25561x+1.07314$	$D_{8}=-0.87237x+1.06154$
$D_{9}=-1.45369x+1.09331$	$D_{10}=-0.23444x+0.87765$

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
