# Peer review of "A Sensitive Frequency Range Method Based on Laser Ultrasounds for Micro-Crack Depth Determination"

_sensors, 2022, doi:10.3390/s22197221_

Round 1
Reviewer 1 Report
The current articles discusses a method for micro-crack measurement using Rayleigh wave transmission. Even though the approach is interesting, the article in its current form requires major updates. Following are specific comments to improve the submission:
- The article includes dashes in words that do not require them throughout the abstract and the different sections of the article.
- The introduction require more background to set the stage for the problem in place.
- The literature review is combined all in a single paragraph which should be split to help the reader evaluate the need for this work.
- The introduction as well as the body of the article has a number of grammatical errors that should be updated. Couple examples: l. 68 “is belonged” should be “belongs”, l. 71 “depth less” should be “depth is less”, l. 90-94 does not make much sense, etc.
- The introduction should not include a summary of the results and method.
- L. 114: the experimental results do not agree with simulations, it should be the opposite.
- L. 124: “frequency domain, The” should be frequency domain. The”
- The math script in the methods section is erroneous.
- L. 158-159: how was the b range calculated?
- Section numbering is inconsistent
- Table 1: which material are the authors using?
- Table 2: is this the depth of the sample or crack? Use more descriptive captions overall in the article.
- Fig. 6 uV should be “\mu V”
- L. 389-391: incorrect figure cross referencing. Fig10 (a) should be (b) and (b) should be (a)
- For the results in figure 12, why does the 0.2 mm crack not follow the same trend of decreasing transmission coefficient? (The authors claim that it does, why?)
- L. 415: what was the sensitive frequency range used for in equation 1?
- The reviewer caught spelling mistakes such as l. 483 “decrees” should be “decrease”.
- The discussion is vague and should be better described in the writing.
Reviewer 2 Report
In this research, to estimate the depth of the micro-surface cracks, waves with different frequencies were transmitted in the cracks, besides the sum of the transmitted coefficients in the frequency domain was used to appraise the surface crack depth. Well-organized presentation of technical literature and figures, use of the FEM method for data analysis and numerical simulation, and the non-destructive proposed method are the robust points of the research. However, the lack of sufficient examination of the results and the existence of many writing problems are available in the current research. I think there are some opaques in the paper that need to be considered as below:
1. A concise introduction and the topic gap should be presented (In the abstract and conclusion section). In addition, in the abstract section, the results and achievements of the research should also be noticed.
2. In keywords, the first character of the word laser should be written with a capital letter (i.e., laser).
3. In the introduction section, the types of methods, advantages, drawbacks, the type of measured parameter, etc., should be presented in tabular form. In this case, it will be more straightforward to follow the text.
4. All parameters in Lines 135-148 (Pages 3 and 4) should be reported in italics. Please do this for all Equations.
5. Please correct the horizontal axis of Figure 3 in terms of writing (put a space after value).
6. On Page 6, Line 214, use “Eq.()” in lieu of “formula”. Lines 216-221 etc., also have the same issue. Please employ the identical format throughout the paper.
7. Please set a number to the Equation on Page 6, Line 228, and present it separately from the text.
8. Page 7, Line 240, to refer to section 3.2, use English numbers, not Greek ones. Similarly, on Page 8, Line 271, etc.
9. In Figures 2, 5, and 6, for better comparison and also for printing purposes, patterned lines should be used instead of solid lines (in addition to applying distinct colors).
10. Page 14, Line 415, what does “fi” mean?
11. Page 16, Lines 439-440 should be re-checked (how to refer to the Figure). This issue exists in other sections as well.
12. Page 16, Line 452-453, it seems referring to Figures are incorrect. Please check.
13. Hyphens are used in many words, such as Page 4, Line 154 (‘fre-quency”), Line 165 (“abil-ity”), Line 176 (“proper-ties”), etc. These items should be corrected in the whole article. In general, there are many typographical errors in the way of referring to Figures, Tables, and References. In addition, there are also grammatical mistakes in the entire paper.
14. The coefficient of Determination (R2) related to Figure 13 (a-e) and Table 3 could be reported. This parameter could be interpreted in the text. Furthermore, please remove the asterisk signs (*) from the entire Table 3.
15. How about investigating the propagation of sound waves through asphalt pavement to assess its health and validate this technique through experimental examination?
16. Did you investigate a given level of deterioration by applying the vibroacoustic impact of vehicular traffic on the pavement to associate an “acoustic signature”?
FYG: The spectral content of an acoustic signal that departed through an undamaged pavement is distinct (different) from that of a signal that traveled across a damaged pavement.
17. In Figure 13 the number of observations are not statistically enough to conclude that “… the simulation results are very well consistent with the experimental data…”. It should be proved via statistical index.
18. Fitting a line to dots (Table 3) in Fgiure 11 suffers from an underfitting error in some cases specially Figure 11(e).
19. The same issue as it was raised in comment number 17 can be mentioned on Figure 14.
20. Have the authors checked other sensors to meaure the crack and compare them like laser scanner, ultra sonic, radar, lidar, ….
Round 2
Reviewer 1 Report
The authors have improved on the article and responded to most comments. Few minor comments are presented here:
- One example of cracks that this work would be applicable to is solidification cracking. A great example that can be included in the introduction was visualized recently by Kouraytem et al. (2021) Additive manufacturing https://www.sciencedirect.com/science/article/pii/S221486042100124X
- l103-104 is unclear, please revise: "Whether by the SLS or the nonlinear modulation method, the potential and extract harm is ignored that the faster expansion of crack is caused by the laser due to the laser heat on the surface crack."
- Fig.6: It is not incorrect but it would be easier to visualize the results if all graphs have the same maximum displayed amplitude. The authors are encouraged to use the same y-axis labels.
Reviewer 2 Report
All comments have been considered accordingly in the paper.
Author Response
Thank you very much for your comments